# HUB: Enhancing Learned Optimizers via Hybrid Update-based Strategy

## Abstract

Learned optimizers are pivotal in meta-learning and recent advancements in scalable learned optimizers have showcased superior performance over traditional, hand-designed counterparts in diverse tasks. However, their adoption is impeded by certain limitations, such as difficulties in handling out-of-distribution tasks, uncontrollable behaviors, and inferior performance in fine-tuning tasks. To address the issue of generalization in these optimizers, we propose a Hybrid-Update-Based (HUB) optimization strategy, inspired by the latest advancements in prompt tuning and result selection techniques in large language and vision models. Compared to previous methodologies(Pr'emont-Schwarz et al., 2022; Heaton et al., 2020), our approach enables a more sophisticated integration between hand-designed and learned optimizers and significantly reduces the computational overhead of hybridization. Our approach broadens the applicability of learned optimizers to tasks beyond their initial training distribution, and it has been validated through a series of diverse tasks, demonstrating significant advantages and unique robustness against out-of-distribution tasks compared to meticulously hyperparameter-tuned competitors. In this paper we also delve into a theoretical analysis of the hybrid strategy's impact on the behaviors and inherent traits of learned optimizers, offering deeper insights into their functionalities and interactions.

## 1 Introduction

Meta-learning, an advancement in AI, seeks to automate the intricate process of research design using deep neural networks. Its spectrum covers hyperparameter tuning (Baik et al., 2020; 2023), architecture search (Zoph & Le, 2016), and initialization strategies for transfer learning (Finn et al., 2017a). Among its applications, learning gradient descent methods stands out (Chen et al., 2016; Andrychowicz et al., 2016; Metz et al., 2020; Harrison et al., 2022; Metz et al., 2022). The Versatile Learned Optimizer (VeLO) (Metz et al., 2022) exemplifies excellence in this domain. It has been trained for over four thousand TPU months, surpassing the previous state-of-the-art model by three orders of magnitude in the number of tasks undertaken during training(Metz et al., 2020). Its extensive training enables it to overcome local minima using generated gradients, displaying impressive convergence speed across a broad array of tasks. For instance, in the benchmark, VeLOdrome (Metz et al., 2022) consists of 83 canonical tasks: (1) On around half of the tasks, VeLO got 4 times faster convergent speed and better results than the learning rate tuned Adam. (2) On more than 14% of the tasks, VeLO is more than 16 times faster. However, like many other neural network-based optimizers, VeLO still struggles with out-of-distribution tasks. Traditional optimizers such as SGD(Gower et al., 2019), RMSProp (Dauphin et al., 2015), and Adam (Kingma & Ba, 2014), rooted in human-designed rules, often show robustness in these situations. These rules ensure convergence in convex optimization problems(Kingma & Ba, 2014; Gower et al., 2019; Pr'emont-Schwarz et al., 2022), while the behavior of the optimizer can be regulated through carefully tuned hyperparameters. In contrast, learned optimizers acquire their rules from extensive data, rendering the optimization process free from hyperparameters but challenging to control. Consequently, even a few misguided steps within this "black box" can significantly disrupt the entire training process, with no means to prevent such adversities. Our experiments further revealed an additional issue not addressed in the original paper: VeLO exhibits poor performance in fine-tuning tasks (see Figure3). We hypothesize that transfer learning aims to preserve pre-trained parameters and sustain high proficiency in acquired knowledge. However, learned optimizers like VeLO treat the model as if it were starting

from scratch, leading to substantial perturbations in pre-trained weights and biases and ultimately resulting in subpar fine-tuning outcomes. The aforementioned discovery further emphasizes the potential disadvantages of relying on hyperparameter-free learned optimizers.

A commonly employed strategy to address the aforementioned issues involves fine-tuning the learned optimizer itself. However, this approach is often considered undesirable for several reasons. First, there is no definitive ground truth for fine-tuning these optimizers, which necessitates relying on meta-training techniques such as reinforcement learning (Li & Malik, 2017) and Evolution Strategies (ES) (Chen et al., 2016; Andrychowicz et al., 2016; Metz et al., 2020; Salimans et al., 2017; Vent, 1975; Nesterov & Spokoiny, 2017). Second, the target space for optimizers is inherently broad, making it challenging to fine-tune optimizer models. Lastly, a critical question arises when faced with disappointing training results: Is it justified to invest time in fine-tuning the optimizer, or would those resources be better allocated towards enhancing the model itself, which ultimately determines its efficacy?

To overcome the challenges of fine-tuning scalable learned optimizers, we propose a strategy inspired by recent advancements in prompt tuning. Prompt tuning techniques leverage input modifications to unlock the latent potential of the model without requiring any alterations to the model itself, this perfectly meets our pursuit.(Wen et al., 2023; Menon & Vondrick, 2022; Gal et al., 2022; Liu et al., 2022; Li & Liang, 2021). Generally speaking, prompt tuning modifies tokenized text or image inputs. However, when it comes to learned optimizers, the inputs usually involve loss, gradient, learned optimizer state, and target model parameters. The complexity of the inputs makes it challenging to directly implement the tuning techniques into the learned optimizer. The previous approach, on the other hand, alternatively uses a hand-designed optimizer and a learned optimizer in each step based on the assessment of future loss. This technique is referred to as LGL2O (Pr'emont-Schwarz et al., 2022). Indeed, LGL2O exemplifies the notion of incorporating human guidance to steer learned optimizers, presenting a viable approach. However, there are several evident drawbacks associated with employing LGL2O: (1) The computational cost would significantly increase (around double), as it requires independent calculation of future losses for two optimizers. (2) During the training of learned optimizers, the objective function is formulated based on meta-loss acquired from multiple tasks. This enables the model to gain global considerations and enhance convergence speed. Consequently, abruptly transitioning between optimizers would undermine the global considerations of the learned optimizer. (3) LGL2O has only been evaluated on vanilla L2O (learning to optimize), which represents a basic form of learned optimizer proposed in (Andrychowicz et al., 2016). Hence, in order to mitigate these drawbacks, we have intelligently incorporated the outputs of VeLO with hand-designed optimizers at each step based on the gradient matrix in every layer (See Figure 1). By doing so, we not only achieve a reduction in computational cost through shared gradient matrices but also effectively harness the capabilities of both learned and hand-designed optimizers in a seamless manner. This approach is referred to as a hybrid-update-based (HUB) optimization strategy.

To summarize, our main contribution can be listed as follows:

- We propose a hybrid approach that effectively integrates the mathematical rules of hand-designed optimizers with learned optimizers. Compared to prior work(Pr'emont-Schwarz et al., 2022; Heaton et al., 2020), our method offers a more refined blending mechanism in each step through the adaptive adjustment of weighting allocation towards both hand-designed and learned optimizers using SoftMax. Additionally, computational overhead is significantly reduced by utilizing a shared gradient matrix.

- We validate the effectiveness of the hybrid strategy across a range of mainstream neural network architectures, including MLP(Strümpler et al., 2021; Yang et al., 2022), CNN(Chollet, 2016; He et al., 2015), RNN(Hasani et al., 2020; Hochreiter & Schmidhuber, 1997), Neural ODE(Hasani et al., 2020; Chen et al., 2018), and Transformer(Dosovitskiy et al., 2020). The assessed tasks involve trajectory fitting, image classification, autonomous driving, and image compression using datasets of different scales. These datasets include the HiP-CT 3D organ image dataset(Walsh et al., 2021), DeePiCar lane-keeping dataset(Bechtel et al., 2018), CIFAR10, CIFAR100(Krizhevsky, 2009), Tiny-imagenet, and Imagenet1K(Deng et al., 2009b) image classification dataset.

- We conducted multiple behavior analysis experiments to analyze the extent to which our hybrid strategy altered VeLO's behavior and showed its robustness when facing out-of-distribution tasks compared to other existing methods.

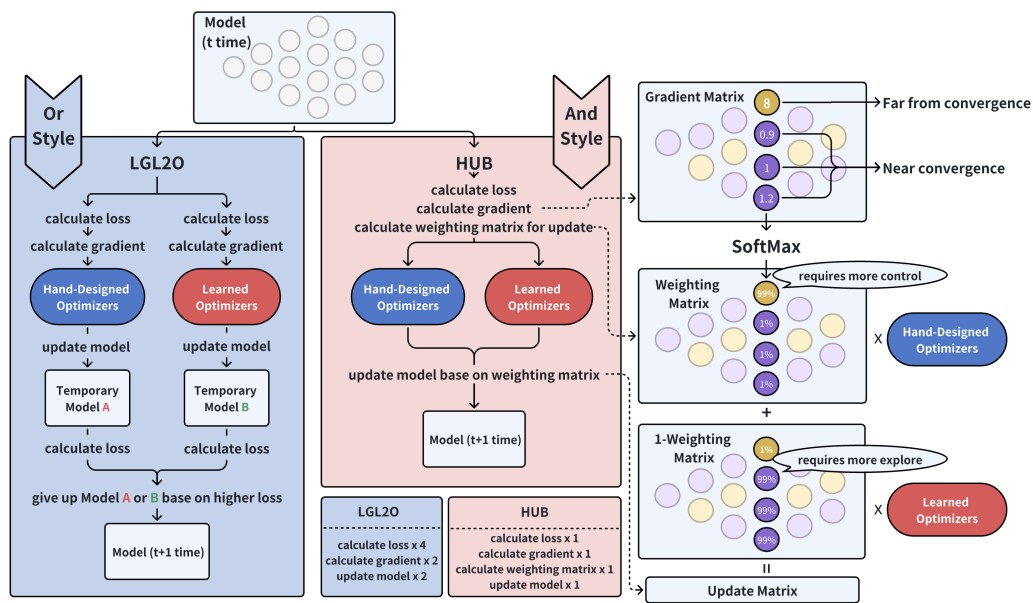

Figure 1: Pipeline of pervious method LGL2O(Pr'emont-Schwarz et al., 2022) and HUB. The LGL2O framework alternatively utilizes a hand-designed optimizer and a learned optimizer, similar to an "or" statement. In contrast, the HUB approach follows an "and" process, resulting in a smoother transition. On the right side of the figure, we demonstrate the rationale behind vanilla HUB. HUB uses a sharing gradient matrix to avoid repetitive loss calculations and with SoftMax, it enables adaptive separation of parameters that are close to convergence from those that are not.

## 2 RELATED WORKS

Conventional hand-designed optimizers, such as Stochastic Gradient Descent (SGD) (Gower et al., 2019), are typically developed by domain experts using a wealth of theoretical knowledge and practical experience. They implement explainable rules like momentum and normalized gradients (Duchi et al., 2011) to refine model parameters and are widely adapted for deep learning problems, spawning various stochastic optimization algorithms like Nesterov (Dozat, 2016), Adagrad(Duchi et al., 2011), Adam, and Adamax (Kingma & Ba, 2014). These first-order optimization algorithms, akin to Recurrent Neural Networks (RNNs) with potential hidden layers, process the gradient of the current parameters and produce updated parameter values at each step. For instance, Nesterov's momentum operates as a hidden layer, Adam uses an exponential average of first- and second-order moments as its hidden layer, and Adamax employs an exponential average of first- and infinite-order moments. These traditional approaches have their set of challenges. They tend to be inefficient in optimizing dynamic systems like RNNs or numerical physics simulations (Metz et al., 2021) due to the problems of gradient vanishing/exploding. Moreover, when faced with non-smooth and non-convex optimization issues, traditional gradient-based methods find it challenging to efficiently reach global optima (Du, 2019) without extensive hyperparameter tuning. Especially in expansive output spaces, they struggle to converge to optimal solutions or critical points in nonconvex landscapes (Reddi et al., 2019).

In contrast, the learned optimizer represents a paradigm shift, using data-driven models to formulate optimization methods and aiming to supersede hand-designed algorithms with neural network-based models (Finn et al., 2017b). This approach encompasses several domains, including black-box optimization (Chen et al., 2017), Bayesian swarm optimization, min-max optimization (Shen et al., 2021), and others. VeLO (Metz et al., 2022) is a prime example of implementing this concept, offering a scalable model that meta-trains using the function value of the current and a fixed number of previous steps as its loss function and employing Evolution Strategies (ES) (Salimans et al., 2017; Vent, 1975; Nesterov & Spokoiny, 2017) for optimization to tackle gradient-related issues. Learned optimizers have shown the ability to converge faster and outperform on trained tasks but

are challenged by tasks that are out-of-distribution and often serve as "black boxes", lacking strategies for effective control. To solve this issue, LGL2O (Pr'emont-Schwarz et al., 2022) outlines a feasible pathway for integrating human rules to guide learned optimizers. However, it is only tested on the very basic tasks and learned optimizer. Our method, tested against state-of-the-art learned optimizers on advanced tasks, substantiates its efficacy and represents a step forward in addressing the limitations of both hand-designed and learned optimization methods.

## 3 METHOD

### 3.1 HUB AS CONTINUOUS TUNING PROCESS

**Overview** In each optimization step, the inputs for the optimizers are derived from the target model. For instance, at time $t$ the inputs includes the gradient of each parameter $g^{(t)}$, the hidden state of the learned optimizer $\theta_L^{(t)}$, and the hidden state of the hand-designed optimizer $\theta_H^{(t)}$. The HUB strategy utilizes gradients as a reference for hybridization and modifies values that need to be updated $U_{HUB}$ on each parameter in the target model:

$$Model(\theta^{(t+1)}) = Model(\theta^{(t)}) + U_{HUB}(g^{(t)}, \theta_H^{(t)}, \theta_L^{(t)}) \tag{1}$$

After assigning modified update values to the target model, subsequent iterations begin with the inputs generated by the current target model being passed to the optimizers, completing a continuous tuning process towards the inputs of the optimizers.

**HUB: Hybrid-Update-Based Optimization Strategy** Learned optimizers differ significantly from hand-designed optimizers in several aspects:

1. Hand-designed optimizers are controllable with hyperparameters and guarantee convergence to the global minimum point with strictly convex and L-smooth objective functions. For first-order algorithms $U_H(g^{(t)}, \theta_H^{(t)})$, this condition is expressed as:

$$\lim_{t \to \infty} g^{(t)} = 0 \Rightarrow \lim_{t \to \infty} U_H\left(g^{(t)}, \theta_H^{(t)}\right) = 0 \tag{2}$$

Full proof of the stability of Adam and Adamax is provided in Section B.2 of the supplementary material.

However, when facing with real world tasks, hand-designed optimizer can get trapped into local minima and find suboptimal stable points with zero gradients during the later stages of optimization, leading to premature stagnation.

2. Learned optimizers exhibit strong adaptability in escaping saddle points and have a better global consideration for optimization, especially when faced with gradient vanishing. They can, to some extent, ignore gradients while still retaining their ability to actively explore. However, this ability is not always beneficial: even if a learned optimizer approaches the global minimum point closely, it may mistakenly believe that it has reached a local minimum and continue exploring other locations. Once it deviates significantly from the true global minimum, subsequent optimization may fail to rediscover it and instead converge to a suboptimal local minimum.

Therefore, a natural idea is to leverage the benefits of both optimizers and address their limitations through mutual guidance. Consider a deep neural network with L layers with $h_l$ being the output of the l-th layer and $\theta_l$ representing the parameters of the l-th layer. For a deep neural network, we can represent the forward computation process as:

$$h_L = f_L \circ f_{L-1} \circ ... \circ f_1(x). \text{ Here, } h_l = f_l(h_{l-1}; \theta_l) \tag{3}$$

During the backpropagation process, we compute the gradient of the loss with respect to $\theta_l$ as follows:

$$\frac{\partial Loss}{\partial \theta_l} = \frac{\partial Loss}{\partial h_L} \prod_l^L \frac{\partial h_{l+1}}{\partial h_l} \frac{\partial h_l}{\partial \theta_l} \tag{4}$$

For activation functions like sigmoid, whose gradient's absolute values are no larger than 1/4, when calculating the product of multiple matrices in the above equation, if a large number of matrix values

are small and the parameters are far from the output, it is easy to encounter the gradient vanishing problem due to the multiplication of a large number of small values. By contrast, for layers closer to the output, the gradient values are relatively large, resulting in significant differences in gradient magnitudes across different layers. Based on this conclusion, we choose to evaluate the gradient value of each parameter within the same layer. Specifically, we apply the SoftMax function to the absolute value of the gradient matrix in each layer $l$ and generate a hybrid reference weighting matrix for the corresponding update matrix (see Figure 1). HUB can be defined as the combination of a hand-designed optimizer $U_H(g^{(t)}, \theta_H^{(t)})$ and a learned optimizer $U_L(g^{(t)}, \theta_L^{(t)})$, expressed by the following equation:

$$U_{HUB}(g^{(t)}, \theta_H^{(t)}, \theta_L^{(t)}) = \sigma(g_l^{(t)}) \odot U_H(g^{(t)}, \theta_H^{(t)}) + (1 - \sigma(g_l^{(t)})) \odot U_L(g^{(t)}, \theta_L^{(t)})$$

For the $i$-th parameter in layer $l$ $\sigma(g_l^{(t)})_i = SoftMax(g_l^{(t)})_i = \dfrac{exp(|g_i^{(t)}|)}{\sum_{j \in layer\ l} exp(|g_j^{(t)}|)}.$ (5)

It is easy to find that with this defualt HUB, most of the weight would be allocated to learned optimizer because of the usage of SoftMax, we would present the proof of this conclusion in Section B.3 of the supplementary material. The rationale behind this approach is to preserve the properties of the learned optimizer and regulate its behavior only for parameters with large gradients using hand-designed optimizers (see Section B.4 for more detail). However, this is not a universal rule. For instance, when facing fine-tuning tasks, we may choose to invert the hybrid reference weighting matrix. Because, unlike training from scratch, here we aim to avoid disturbing pre-trained weights and biases, we rely on learned optimizers to fast finetune for parameters with higher gradients and set a small learning rate for hand-designed optimizers to carefully finetune for parameters with small gradients in this case. We will discuss more about the variations of HUB and their suitable scenario in Section C of the supplementary material.

## 3.2 INVESTIGATION OF HUB BEHAVIOUR

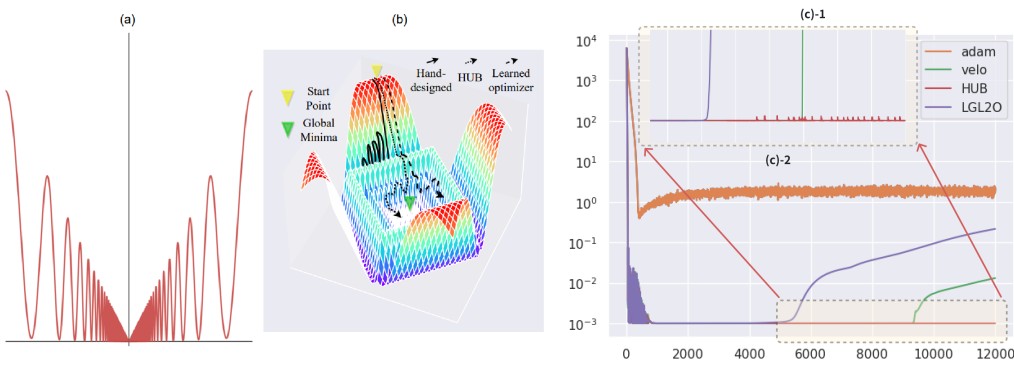

Figure 2: Theoretical optimization experiment. (a) illustrates the experimental function on a 2D coordinate, showcasing its characteristics. (b) demonstrates the descending paths taken by different optimizer, with HUB selecting a relatively superior path that converges to the lowest loss. (c)-1&2 displays the loss curve, indicating that HUB is the only method that successfully discovers and remains close to the global minimum.

As the learned optimizer is considered a black box solver, we conduct an optimization experiment to examine the impact of HUB. The following continuous function has an explicit global minimum point at 0 and exhibits convexity away from this point:

$$f(x) = \begin{cases} 0 & x = 0 \\ \|x\|_1(1 + \lambda\|x\|_1 + \cos\frac{1}{\|x\|_1}) & otherwise. \end{cases}$$

Here, $\lambda = 0.01, d = 1000, x \in \mathbb{R}^d, \|x\|_1 = \sum_i |x_i|.$ (6)

When $x$ is large, the function approximates a convex function:

$$\cos \frac{1}{\|x\|_1} = O(1) = o(\lambda\|x\|_1), f(x) \approx \|x\|_1(1 + \lambda\|x\|_1)(x \to \infty).$$

However, when approaching global minima, the gradient undergoes significant changes and displays non-convexity (see Figure 2(a)).

In the initial training phase, HUB, LGL2O, and VeLO exhibit a more pronounced downward trend in comparison to hand-crafted optimizers. Although this function does not represent a real-world neural network, learned optimizers still demonstrate superior global perspectives on this function due to its resemblance to neural networks, resulting in a swift reduction of loss.

During the intermediate stage of training, significant changes in loss are experienced by all methods, revealing a more intricate non-convex scenario. Nevertheless, VeLO, LGL2O, and HUB adeptly navigate these challenges, while Adam becomes ensnared in suboptimal stable points, continuing to oscillate. It's critical to clarify that this oscillation does not denote the absence of stability. Instead, it reflects substantial fluctuations in function values, even around less optimal stability points, as depicted in Figure 2(b).

In the later stages of training, VeLO and LGL2O undergo a swift augmentation in loss, signaling potential instability which can adversely affect the optimization process. Despite having discovered stable points, VeLO tends to prioritize further exploration, risking suboptimal results. A similar scenario is observed with LGL2O; it doesn't choose the hand-designed optimizer as long as it predicts a higher future loss, compromising the ability to control the learned optimizer behavior (see Figure 2(c)-1). Conversely, a close examination of the HUB method reveals that its curve manifests multiple minor fluctuations post 7500 steps. This suggests that although HUB also encounters abrupt gradient alterations, it swiftly reverts to the vicinity of the preceding local minimum. HUB's curve is indicative of a negative feedback regulation mechanism; it assigns augmented weight to the hand-designed optimizer following abrupt gradient changes, efficiently counterbalancing VeLO's erroneous trajectory along the negative gradient axis and thus sustains stable performance (see Figure 2(c)-2).

## 4 EXPERIMENTAL EVALUATION

In this section, we present the experiments conducted to validate the effectiveness of HUB across various mainstream neural network architectures. Detailed setups and results, including model size, model hyperparameters, optimizer hyperparameters, training pipeline, and tuning-related explanations, can be found in Section A of the supplementary material.

### 4.1 OUT-OF-DISTRIBUTION TASKS

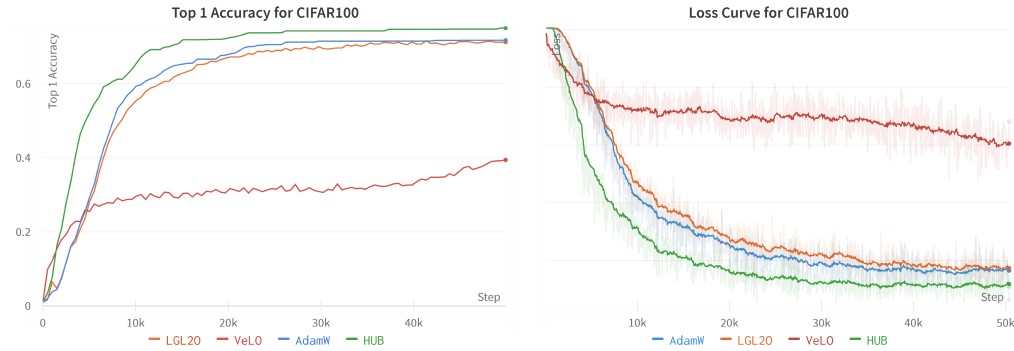

Figure 3: Top 1 accuracy (Left) and the loss curve (Right) for pre-trained Xception on CIFAR100.

**Fully Fine-tune Task** Learned optimizers, like VeLO, face difficulties in distinguishing between train-from-scratch tasks and fine-tuning tasks. We employ the Xception model (Chollet, 2016) as our pre-trained model, utilizing the Imagenet1K dataset for pre-training.

Table 1: Evaluating Imagenet pre-trained Xception on downstream image classification task. HUB not only surpasses VeLO but also achieves superior performance compared to heavily tuned AdamW.

| Evaluation Metric:Top1/Top5 Accuracy ↑ | | | | |
|---|---|---|---|---|
| Dataset | CIFAR10 | CIFAR100 | Tiny-imagenet | Total Runtime↓ |
| AdamW | 89.24%/99.54% | 73.71%/92.64% | 56.23%/73.62% | **17h42m** |
| VeLO | 72.60%/96.87% | 40.05%/60.33% | 16.45%/25.40% | 18h25m |
| LGL2O | 89.01%/99.19% | 73.12%/92.36% | 50.30%/67.9% | 38h12m |
| HUB | **89.79%/99.64%** | **75.20%/94.20%** | **59.16%/74.00%** | 19h09m |

For the downstream tasks, we use the CIFAR10, CIFAR100, and Tiny-imagine datasets. Adamax is hyperparameter-tuned and we also collect the total runtime for 3 training tasks on an A100GPU. A detailed setup can be found in Section A.1.

**Unseen Network Architecture** We performed classical trajectory-fitting tasks and lane-keeping tasks using an out-of-distribution Liquid Time Constant (LTC)(Hasani et al., 2020) architecture. The LTC architecture incorporates the Neural

Table 2: LTC on lane-keeping and trajectory-fitting tasks.

| Model:LTC Metric:MSE↓ | | | | |
|---|---|---|---|---|
| Optimizer | Adam | VeLO | LGL2O | HUB |
| Trajectory-fitting | 0.0119 | NAN | 0.1641 | **4.8663e-5** |
| Lane-keeping | 9.1376 | 5378.1544 | 78.1195 | **3.1201** |

ODE structure into a recurrent process to enhance neuron expressivity. (see Section A.1 in the supplementary material for a detailed introduction)

We followed the experimental setups reported in prior works (Hasani et al., 2020; Lechner et al., 2020), using a 4-layer, 8-cell LTC for sine curve fitting and a 19-cell LTC for lane-keeping task. Adam was used as a hyperparameter-tuned optimizer. The reference code-base and hyperparameter selection were based on the project demonstration[1].

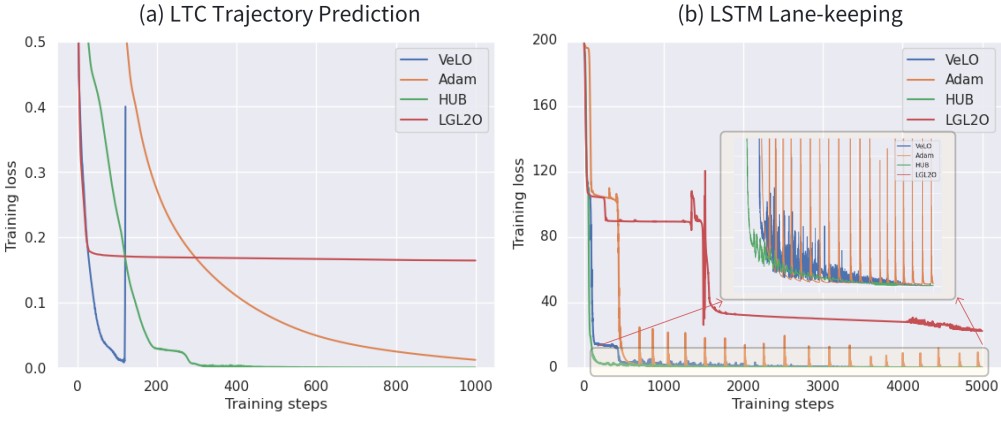

Figure 4: Loss curve of LTC trajectory-fitting(a) and LSTM lane-keeping(b) tasks. In scenario (a), VeLO experienced a gradient explosion and ultimately failed. In scenario (b), Adam also encountered gradient fluctuations. However, HUB demonstrated its robustness in both scenarios and outperformed other competitors.

Figure 4(a) illustrates several noteworthy patterns amongst the optimization algorithms. Initially, VeLO demonstrates a swift decrease in loss but ultimately fails to converge due to a gradient explo-

---

[1]Codebase: `https://github.com/mlech261/ncps`

sion. LGL2O, leveraging VeLO's rapid initial loss drop, initially experiences expedited convergence but eventually becomes ensnared in a suboptimal point due to VeLO's inherent failures. Conversely, both HUB and Adam exhibit convergence; however, HUB uniquely capitalizes on the rapid initial loss drop of VeLO and the superior generalization ability of Adam, resulting in faster convergence and improved outcomes.

## 4.2 IN DISTRIBUTION TASKS

**Autonomous Driving with RNN** We assessed our approach using gradient-sensitive network structures, employing an LSTM model(Hochreiter & Schmidhuber, 1997) with 64 hidden units on the DeepPiCar lane-keeping task, with Adam serving as a hyperparameter-tuned optimizer. The runtime was also calculated for a total of 5,000

Table 3: LSTM on lane-keeping tasks.

| Task: Lane-keeping with LSTM | | | | |
|---|---|---|---|---|
| Optimizer | Adam | VeLO | LGL2O | HUB |
| MSE↓ | 13.3561 | 8.2062 | 25.4467 | **4.2675** |
| Runtime↓ | **578s** | 649s | 1205s | 698s |

training steps to gauge the difference in computational overhead between the various methods. (See Section A.1 for more detailed setups.)

From the observed loss curves, it is evident that Adam experienced substantial fluctuations, suggesting a significant impact from unstable gradient flows. VeLO exhibited robustness in this scenario, maintaining stability amid the unstable gradients. However, unexpectedly, LGL2O failed to converge to a satisfactory value. We discovered that LGL2O continuously oscillated between Adamax and VeLO during expected scheduled loss drops. This erratic behavior resulted in LGL2O missing opportunities to stabilize and converge. On the contrary, HUB demonstrated stable and swift convergence in the initial training phase and maintained robustness in the later stages. Moreover, while LGL2O requires around double the computational resources, HUB managed to mitigate this overhead to approximately 10-15%

**Image Compression with Implict Nueral Network** We conducted the image compression task on HiP-CT 3D organ image dataset(Walsh et al., 2021) with the state-of-the-art Siren MLP model(Strümpler et al., 2021; Sitzmann et al., 2020). This task is prone to overfitting, meaning that achieving better-decompressed quality involves strongly overfitting to the target image. It requires the optimizer to possess the ability to bypass local minima and reach lower loss values. We maintained consistent model and optimizer configurations in accordance with our reference source (Yang et al., 2022). We compressed

Table 4: Performance comparison of MLP (Siren) on the HiP-CT 3D organ image compression task. Peak Signal-to-Noise Ratio (PSNR) is commonly used as a quality measurement between two images, where higher PSNR indicates better absolute quality of the decompressed or reconstructed image.

| Evaluation Metric: Decompressed PSNR ↑ | | | |
|---|---|---|---|
| Image Size | $64^3$ | $256^3$ | $512^3$ |
| Adamax | 48.0427 | 52.2213 | 47.1590 |
| VeLO | 49.5022 | 52.6720 | 47.8934 |
| LGL2O | 49.6106 | 52.5798 | 47.9298 |
| HUB | **50.1202** | **53.2311** | **48.3011** |

multiple images at resolutions of $64^3$, $256^3$, and $512^3$, including brain, kidney, heart, and lung. Figure 5 demonstrates that the LGL2O loss curve almost overlaps with the VeLO loss curve, indicating a rare utilization of Adamax. HUB initially experiences a slower descent influenced by Adamax's training schedule. Later on, it leverages VeLO's ability to navigate saddle points and achieves lower loss with higher PSNR. (See Section A.1 for more detailed setups.)

**Image Classification with CNN and Vision Transformer** We conducted extensive image classification experiments, utilizing both the Resnet-50 model (He et al., 2015) and the ViT model (Dosovitskiy et al., 2020), on datasets including CIFAR10, CIFAR100, Tiny-imagenet, and Imagenet1K. Our experiments were guided by the specifications outlined in (Lee et al., 2021), encompassing model size, batch size, epoch count, optimizer type, base learning rate, warmup period, and

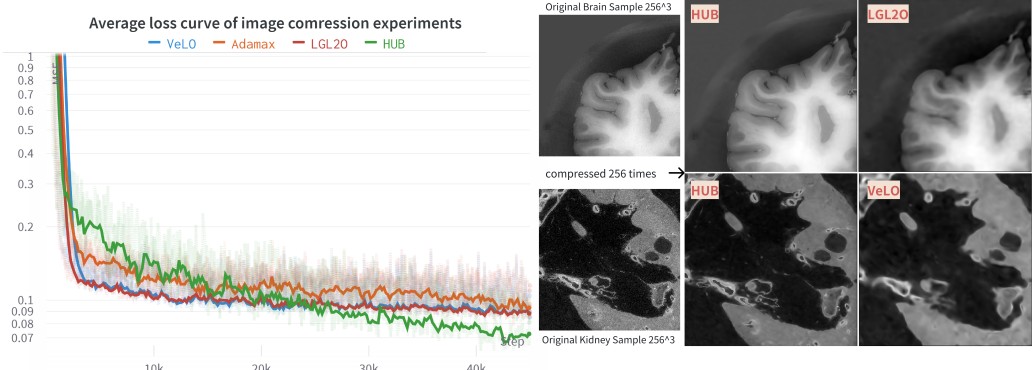

Figure 5: Training loss curves and decompressed images for the MLP image compression task.

Table 5: Performance Comparison on Image Classification Tasks

| Model | Resnet-50 | | ViT-B | | | ViT-L |
|---|---|---|---|---|---|---|
| Dataset | CIFAR10 | CIFAR100 | CIFAR10 | CIFAR100 | Tiny-Imagenet | Imagenet1K |
| Metric | Top1/Top5 Accuracy ↑ | | | | | |
| Adam | 83.06%/99.24% | 73.57%/89.17% | 76.59%/96.49% | 52.96%/75.50% | 44.22%/65.75% | 79.93%/90.52% |
| AdamW | 83.61%/99.13% | 74.07%/90.20% | 84.17%/98.58% | 64.06%/88.56% | 54.30%/71.87% | 82.05%/92.01% |
| VeLO | 83.52%/99.30% | 72.71%/88.74% | 87.22%/99.60% | 69.67%/93.17% | 56.26%/73.37% | 82.65%/92.44% |
| LGL2O | 83.77%/99.33% | 74.71%/88.94% | 88.01%/99.64% | 69.89%/93.44% | 57.07%/74.21% | 82.96%/92.57% |
| HUB | **84.06%/99.42%** | **75.34%/89.60%** | **88.46%/99.79%** | **70.36%/93.96%** | **57.82%/74.94%** | **83.28%/92.93%** |

scheduler, with Adam configured as the baseline for a consistent training setup. For experiments conducted on CIFAR10 and CIFAR100, our standard configuration included a base learning rate of 0.00075, a cosine scheduler, and a warmup for 1/10 of the total epochs.(See Section A.1 and Section A.1 for more detail.) Results, as presented in Table 5, reveal that meticulously tuned hand-designed optimizers are capable of surpassing the hyperparameter-free VeLO optimizer. However, the benefits intrinsic to hand-designed optimizers can be conferred to VeLO via our hybrid method, HUB, effectuating enhanced performance.

The ViT model, noted for its challenges in training from scratch with smaller datasets due to overfitting, was also subjected to our comprehensive investigation. While a plethora of studies emphasizes the presentation of the loss curve to manifest faster convergence and reduced loss, some research suggests that averting model overconfidence in reasonable, albeit not necessarily low, loss zones can augment test performance (Ishida et al., 2020). In this case, we adopt the setups discussed in (Lee et al., 2021), which provide detailed guidelines for training ViT on small datasets.

## 5 FUTURE WORKS, LIMITATIONS AND CONCLUSION

Learned optimizer would be ideal if it could be trained on "sufficient" data, however, this is impractical in reality. Even VeLO's training is limited by resources. In such cases, hand-designed optimizers can serve as an effective inductive bias designed by humans and a potential avenue for future research is to explore how learned optimizers can effectively leverage prior knowledge encoded in optimization rules designed by humans. While our work proposes a non-training solution to this question, we anticipate that more robust solutions may be achieved through guided training of the learned optimizer. In conclusion, in this paper, we proposed a more advanced and efficient hybrid method that allows a learned optimizer to enhance robustness and effectiveness without requiring modifications to the model itself. This aligns with similar conclusions drawn from prior work such as LGL2O, as well as research in other domains, such as computer vision and natural language processing.

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

# A EXPERIMENTS

## A.1 EXPERIMENTAL SETUPS FOR SECTION 4

We present the experimental setup and extra corresponding results (Figure 10, Figure 11 and Figure 12) for the tasks outlined in Tables 1, 2, 3, 4, and 5 in Section 4 of our paper. All experiments were conducted using Haiku, a JAX-based framework.

**MLP**   The Sinusoidal Representation Networks (Siren)(Strümpler et al., 2021) is a multi-layer perceptron model that utilizes the Implicit Neural Representation (INR) method. Siren takes in the corresponding coordinate values (x, y, and z) of the target image as input and outputs its RGB/grayscale value at that specific coordinate. Another well-known INR-based model for new view synthesis is Neural Radiance Fields (NeRF)(Mildenhall et al., 2020).

The Siren network we employ consists of 7 layers, with a frequency rate (fr) of 2.2 and a sinusoidal frequency hyperparameter (w0) set to 20. We utilize the Adam and Adamax optimizers with an initial learning rate of 0.001, beta1 = 0.9, beta2 = 0.999, epsilon = 1e-8, employing cosine learning rate decay without warmup - consistent with our reference codebase(Yang et al., 2022). For VeLO, we use default settings as it is hyperparameter-free; for HUB, we hybridize Adamax (using the same setting as when using Adamax alone) with VeLO. The batch size we use is 100k coordinates. We utilized the HiP-CT dataset(Walsh et al., 2021) as our primary image source, which offers cellular-level imaging of various organisms across multiple anatomical planes. Specifically, we focused on four available organs in this dataset (Lung, Heart, Kidney, and Brain). For the purpose of a standardized and equitable comparison, we have partitioned the data into $64^3$, $256^3$ and $512^3$ sizes, and the raw data was utilized without pre-processing, but the coordinates were normalized to $[-1, 1]^3$ in INR-based methods. We run the experiment on a GeForce RTX3090Ti GPU and the PSNR values presented in Table 4 represent the average of 10 trials for each size, with a compression ratio of x256.

**Resnet**   We use the Resnet-50(He et al., 2015) model in this experiment, the blocks per group are 3, 4, 6, and 3 and the channels per group are 256, 512, 1024, and 2048. The bottleneck is adopted and the stride is (1, 2, 2, 2). For data augmentation, we first resized the training image using the BICUBIC method to (384, 384) and applied Auto Augment(Cubuk et al., 2019), random horizontal/vertical flip and random crop to get the final resolution (224, 224). The batch size we choose is 128 and in total 100 epochs of training. Adam and Adamax optimizer used an initial learning rate of 7.5e-4, with a 10 epochs warmup followed by cosine decay in learning rate. The rationality behind this learning rate is shown in Figure 6. We ran the experiment on an A100 GPU.

**RNN and Neural ODE**   Our experiments focused on the application of RNN and Neural ODE in two specific tasks: trajectory prediction and lane-keeping. For these tasks, we employed three different models. Firstly, we used a long short-term memory (LSTM)(Hochreiter & Schmidhuber, 1997) network with 64 hidden units for both trajectory prediction and lane-keeping tasks. Secondly, we utilized an 8-cell liquid time-constant (LTC)(Hasani et al., 2020) network for the trajectory prediction task, and finally, a 19-cell LTC for the lane-keeping task. LTC models are wired with neural circuit policies (NCP)(Lechner et al., 2020) with default 75% sparsity. These model choices were based on the experiment in (Hasani et al., 2020; Lechner et al., 2020).

LTC represents a novel class of time-continuous recurrent neural network models. Rather than defining a learning system's dynamics through implicit nonlinearities, LTC constructs networks of linear first-order dynamical systems modulated by nonlinear interlinked gates, drawing inspiration from principles of brain-neural computation. Specifically, the neural dynamics of a single LTC cell are governed by continuous-time ordinary differential equations (ODEs) originally developed to capture the dynamics of the nervous system in small organisms like C. elegans. The ODE dynamics of the LTC cell are then integrated into a recurrent process to establish a temporal dimension. This design significantly enhances the expressive power of an LTC cell while increasing its interpretability. The amplification of a single cell suggests that complex tasks can be accomplished using a much smaller LTC model compared to other modern RNN models. Furthermore, the connectivity of LTC cells is defined by NCP, which draws inspiration from biological systems. NCP incorporates a four-layer hierarchical network topology comprising sensory, inter-neuron, command, and motor layers,

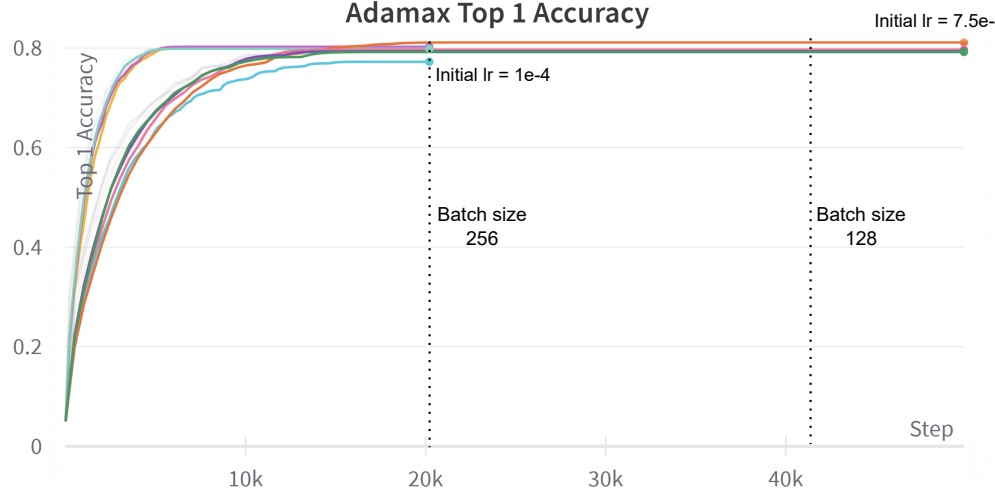

Figure 6: Best top 1 Accuracy for AdamW. This figure demonstrates the tuning process for AdamW, the best top 1 accuracy is the average on CIFAR10 and CIFAR100

along with polarity (inhibitory and excitatory) connections. This NCP design allows LTC models to achieve a connection sparsity of up to 90% without sacrificing their expressive power.

For the trajectory prediction task, our objective is to predict a sine curve signal wave. During training, three sine curves with different frequencies are used as inputs, and the output is a periodical function curve. This task represents a classical Neural ODE scenario that parameterizes the Fourier transform process. Due to the relative simplicity of the task, we set the learning rates for Adam to 0.01, with beta1 = 0.9, beta2 = 0.999, epsilon = 1e-8, and momentum = 0.9. As for the HUB and LGL2O optimizer, we combine Adam and VeLO without modifying the default hyperparameters.

For the lane-keeping task, we utilize the DeepPiCar dataset, which consists of high-quality indoor close environment images and corresponding turning angles captured by a robotic car. We split the dataset into training, validation, and test sets in a ratio of 75:10:15. Before feeding the images into the LSTM/LTC prediction head, we extract kernel features using a CNN with the structure present in Table 6. Adam is empirically tuned with an initial learning rate of 0.01, which decays in each of the 300 training steps with gamma = 0.3. Similar to before, we combine Adam and VeLO for the HUB and LGL2O optimizer, maintaining the tuned hyperparameters.

The aforementioned task design, data preprocessing, and optimizer setup are based on (Hasani et al., 2020; Lechner et al., 2020), we simply shrink the CNN scale to match up a smaller dataset we use for the lane-keeping experiment. The results in Table 2 are the average of 10 trials. We ran the experiment on a Geforce RTX3090Ti GPU.

**Vision Transformer**   Training Vision Transformer(Dosovitskiy et al., 2020) (ViT) on small datasets can be challenging, as the model tends to strongly overfit the training data without careful selection of hyperparameters and model size, leading to underperformance. In this case, we adopt the setups discussed in (Lee et al., 2021), which provide detailed guidelines for training ViT on small datasets.

Regarding the model size, we configure the ViT with a depth of 9, a hidden dimension of 192, and 12 attention heads. The patch size for the patch embedding layer is set to 8. For image preprocessing, we employ techniques such as CutMix(Yun et al., 2019), Auto Augment(Cubuk et al., 2019), random horizontal/vertical flip, and random crop.

To optimize the model, we utilize AdamW(Kingma & Ba, 2014) as the tuned optimizer. The initial learning rate is set to 0.003, with a warmup period of 1/10 of the total epochs, followed by cosine decay in the learning rate. Adam shares the same hyperparameter settings as AdamW and are used

Table 6: CNN block structure. This CNN block is for feature extraction, so the size is restricted. The CNN structure is similar to (Hasani et al., 2020; Lechner et al., 2020)

| CNN block structure | | |
|---|---|---|
| Layer(type) | Output Shape | Parameter Count |
| conv2d 1 (Conv2D) | (None, 31, 98, 24) | 1824 |
| conv2d 2 (Conv2D) | (None, 14, 47, 36) | 21636 |
| conv2d 3 (Conv2D) | (None, 5, 22, 48) | 43248 |
| conv2d 4 (Conv2D) | (None, 3, 20, 64) | 27712 |
| dropout 1 (Dropout) | (None, 3, 20, 64) | 0 |
| conv2d 5 (Conv2D) | (None, 1, 18, 64) | 36928 |
| flatten 1 (Flatten) | (None, 1152) | 0 |
| dropout 2 (Dropout) | (None, 1152) | 0 |
| dense 1 (Dense) | (None, 100) | 115300 |
| dense 2 (Dense) | (None, 50) | 5050 |
| Total Parameter Count: 251,698 | | |

as baselines. Additionally, we hybridize AdamW with VeLO, incorporating tuned hyperparameters. A weight decay of 0.05 and a batch size of 128 are employed. The training is conducted for a total of 100 epochs on an A100 GPU.

As a validation for scaling up, we followed the instructions outlined in [2] to train a large vision transformer with 16 as the patch size on the Imagenet dataset. The initial learning rate for both AdamW and Adam was set at 0.01, with a total of 20k training steps. Additionally, we implemented a warmup period of 1/10 and cosine decay. For HUB and LGL2O, we utilized hybrid AdamW with VeLO. Other procedures, such as image preprocessing, remain consistent with the aforementioned setups.

**Xception** The Xception(Chollet, 2016) we employ has been pre-trained on the Imagenet1k dataset(Deng et al., 2009a). For our downstream tasks, we selected CIFAR10, CIFAR100, and Tiny-imagenet datasets. Following the methodology outlined in our reference source [3], we conducted full fine-tuning experiments.

Regarding image preprocessing, we initially resized the training images using the BICUBIC method to a resolution of (384, 384). We then applied Auto Augment(Cubuk et al., 2019), random horizontal/vertical flip, and random crop to obtain a final resolution of (299, 299). The batch size utilized during training was set to 128, and we fine-tuned the model for a total of 100 epochs on an A100 GPU.

In this experiment, the hyperparameters for AdamW were carefully tuned. Since it is a fine-tuning task, we empirically determined that a relatively small learning rate is desirable. We found that learning rates below 5e-4 generally yielded better results (see Figure 7). Therefore, we selected an initial learning rate of 1e-4. We employed a warmup period of 1/10 of the total epochs, followed by cosine decay, which is a standard setup.

As mentioned in Section 1 of our paper, VeLO performs poorly in fine-tuning tasks. In Section 3.1, we briefly discussed the reasons behind invert weighting HUB. In this particular case, although we chose to hybridize AdamW with VeLO, the HUB strategy we employed differed from that in train-from-scratch tasks. We will provide detailed explanations of the variations of HUB in the upcoming Section C.

### A.2 ADDITIONAL EXPERIMENTS

**Compare HUB to reinitialize extend training method** Although larger optimizers may achieve satisfactory performance with fewer iterations, it is crucial to consider the potential increase in com-

---

[2]https://github.com/google-research/vision_transformer
[3]Codebase: https://github.com/abarcel/haikumodels

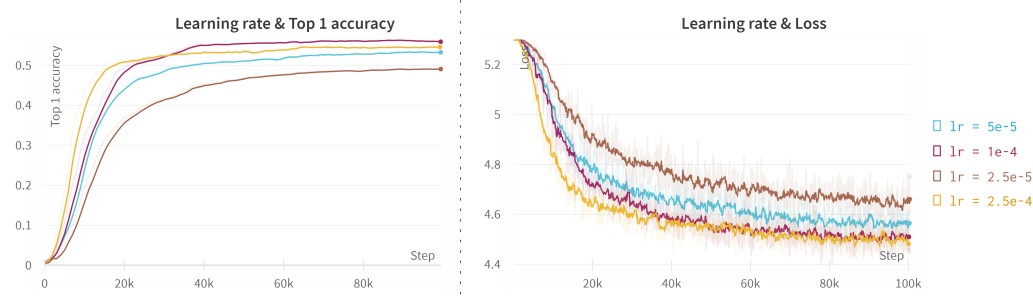

Figure 7: The loss curve and top 1 accuracy (here demonstrate Tiny-imagenet dataset) are closely tied to the initial learning rate. As depicted in the figure, a higher initial learning rate (2.5e-4) leads to faster convergence but results in an early plateau, while lower initial learning rates (5e-5 and 2.5e-5) converge slowly but at relatively suboptimal points. Therefore, an optimal initial learning rate of 1e-4 is recommended.

putational overhead per step. This overhead can result in issues where the gradient and loss values become unstable, thereby impeding the training process. To tackle this challenge, two reinitialization training strategies have been previously proposed in (Metz et al., 2022) and will be compared to the HUB method in this analysis.

The experiment employed in this study is the LTC trajectory prediction task. In our paper, we addressed the issue of NAN when using VeLO as an optimizer. (see section 4.1 RNN and Neural ODE paragraph)

- Increase Steps: Continue from the final optimizer state of the previous run but increase the number of steps.
- Min-Loss Reinit: Continue from the min-Loss optimizer state of the previous run.

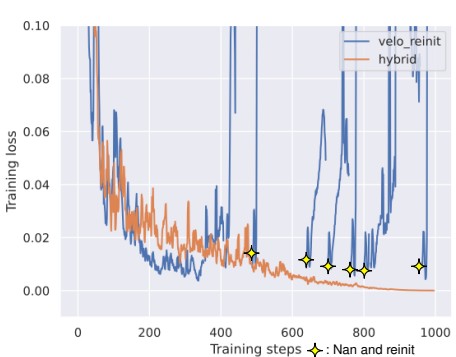
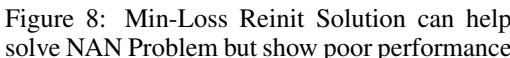

Figure 8: Min-Loss Reinit Solution can help solve NAN Problem but show poor performance

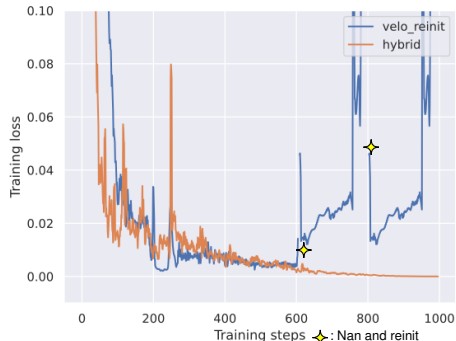

Figure 9: Increase Steps Solution also can help solve NAN Problem but built-in cycle in velo leads to this just repeating the poor performance

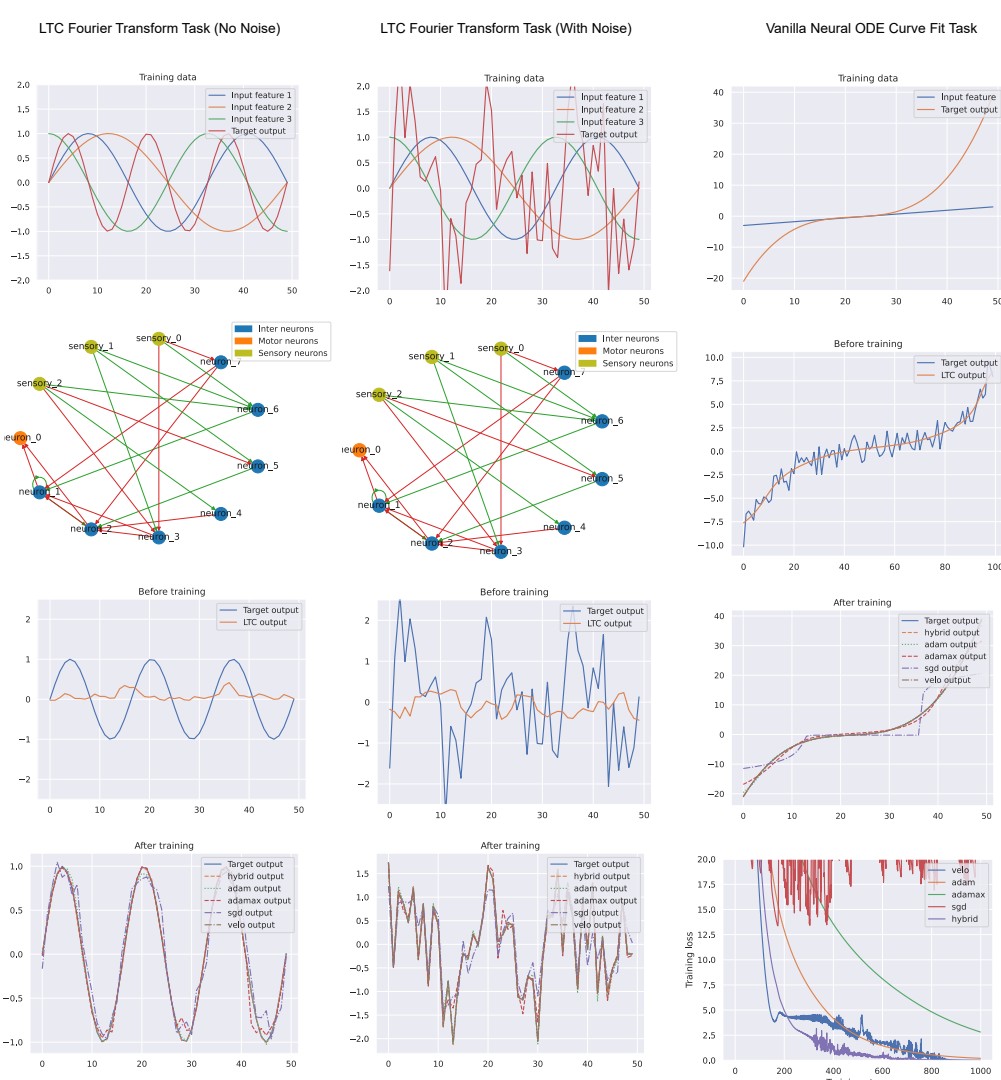

Figure 10: LTC and Neural ODE related trajectory predicting tasks.

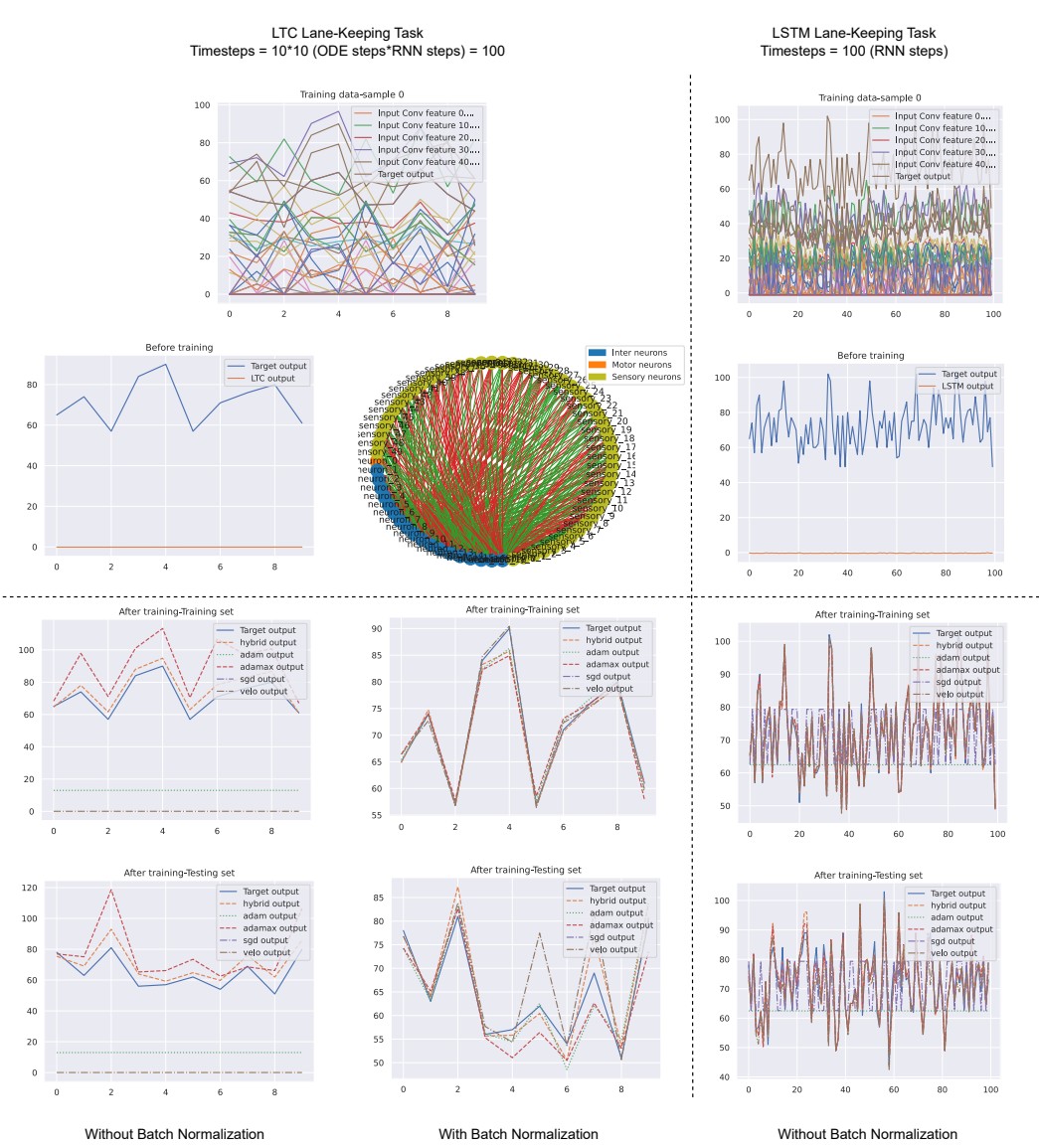

Figure 11: LTC and LSTM lane-keeping tasks.

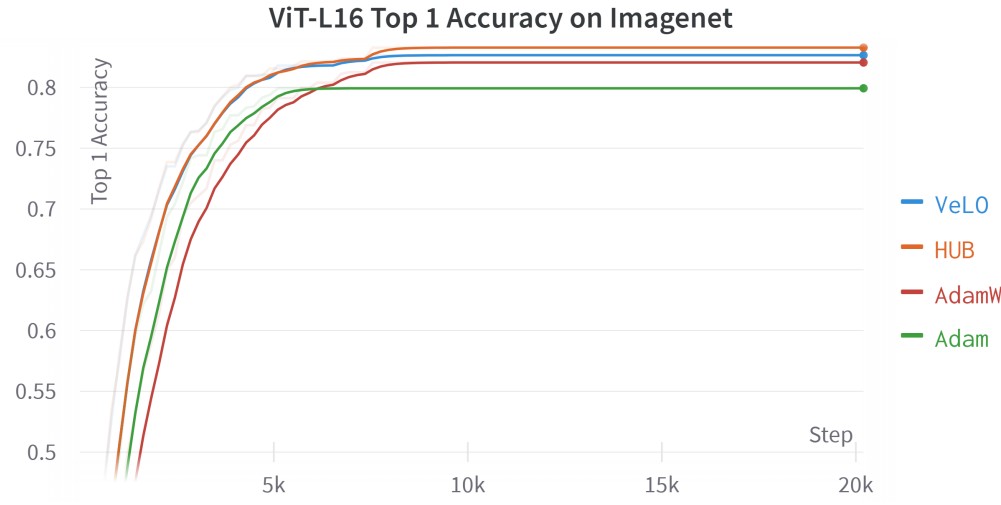

Figure 12: Vision transformer large top 1 Accuracy on Imagenet

## B   THEOREMS AND PROOFS FOR SECTION 3

### B.1   REPRESENTATION FORMAT FOR ADAM AND ADAMAX

With $S^{(t)}$, $M^{(t)}$ and $n^{(t)}$ in the second row representing the hidden layers of Adam and Adamax at time $t$, we have the following RNN-form representations:

$$U_{Adam}(g^{(t)}, \beta_1, \beta_2, \alpha) = \frac{\alpha}{\sqrt{\frac{M^{(t)}}{1-\beta_2^t}} + \epsilon} \odot \frac{S^{(t)}}{1 - \beta_1^t}$$

$$\text{Adam: } S^{(t)} = \beta_1 S^{(t-1)} + (1 - \beta_1)g^{(t)}; M^{(t)} = \beta_2 M^{(t-1)} + (1 - \beta_2)g^{(t)} \odot g^{(t)} \quad (7)$$

$$U_{Adamax}(g^{(t)}, \beta_1, \beta_2, \alpha) = \frac{\alpha}{n^{(t)} + \epsilon} \odot \frac{S^{(t)}}{1 - \beta_1^t}$$

$$\text{Adamax: } S^{(t)} = \beta_1 S^{(t-1)} + (1 - \beta_1)g^{(t)}; n^{(t)} = \max(\beta_2 * n^{(t-1)}, |g^{(t)}|)$$

### B.2   PROOF OF ADAM AND ADAMAX STABILITY

If $\lim_{t \to \infty} g^{(t)} = 0$, assume for any $\epsilon > 0$, there exist time t > T we have $\|g^{(t)}\| < \epsilon$, then:

$$\|S^{(t)}\| = \|\beta_1^{t-T} S^{(T)} + (1 - \beta_1) \sum_{i=T+1}^{t} \beta_1^{t-i} g^{(i)}\| \le \beta_1^{t-T}\|S^{(T)}\| + (1 - \beta_1^{t-T})\epsilon \quad (8)$$

So $\lim_{t \to \infty} \left\|S^{(t)}\right\| = 0$. Similarly $\lim_{t \to \infty} \left\|M^{(t)}\right\| = 0$. Thus $\lim_{t \to \infty} U_{Adam}\left(g^{(t)}, \theta^{(t)}\right) = 0$

When the assumption in equation (7) holds true, Adam and Adamax exhibit good stability properties for strictly convex problems.

### B.3   THE ANALYSIS OF MAIN DEPENDENCIES IN HUB

With the definition of HUB provided in Section 3.1, we can demonstrate its main dependence as follows:

$$\sigma(g^{(t)})_i = \frac{exp(|g_i^{(t)}|)}{\sum_{j \in layer\ l} exp(|g_j^{(t)}|)}, \text{if } |g_i^{(t)}| \gg |g_{k \neq i \in layer\ l}^{(t)}| :$$

$$exp(|g_i^{(t)}|) \approx \sum_{j \in layer\ l} exp(|g_j^{(t)}|) \Rightarrow \sigma(g^{(t)})_i \approx 1. \quad (9)$$

$$\sigma(g^{(t)})_{k \neq i \in layer\ l} \approx 0 \Rightarrow \text{Majority of parameters rely on } U_L(g^{(t)}, \theta_L^{(t)}.)$$

Equation (4) indicates that the HUB strategy exhibits a stronger reliance on the learned optimizer. This trend increases as the layer size grows:

If $|g_m^{(t)}| = |g_n^{(t)}|$ holds for arbitrary m and n in layer $l$ with K parameters,

$$\sigma(g^{(t)})_i = \frac{exp(|g_i^{(t)}|)}{\sum_{j \in layer\ l} exp(|g_j^{(t)}|)} = \frac{1}{K}. \quad (10)$$

### B.4   RATIONALE BEHIND HUB APPROACH

Assumption 1: Consider an optimization scenario where $x^*$ is a local optimum of the continuous loss function $f(x)$. We assume that $f(x)$ is strictly convex and L-smooth. We have $\nabla f(x^*) = 0$, because the gradient at the local optimum point $x^*$ vanishes according to this objective function.

Assumption 2: Section B.2 demonstrates the stability of hand-designed optimizers like Adam and Adamax in converging to local optima in the Assumption 1 scenario.

Considering the black-box nature of the learned optimizer, we discuss its behavior based on two cases:

- Case 1: When almost all the gradients are small and approaching convergence near $x^*$. This scenario resembles the situation depicted in Section B.3, where approximately $\frac{1}{K}$ of the weight for each parameter would rely on a hand-designed optimizer. Here, $K$ represents the number of parameters in that layer (typically large, leading to the dominant role of the learned optimizer).

    - Sub-Case 1: If the learned optimizer is stable: Both optimizers progressively approach zero distance between the current location and $x^*$ as the iteration number $t$ increases.
    - Sub-Case 2: If the learned optimizer is unstable: In the following iterations, adverse optimization along the $m$ dimensions emerges, causing these parameters to deviate from the optimal point $x^*$. This phenomenon subsequently gives rise to Case 2.

- Case 2: When gradients are small and near convergence for some parameters, while others ($m$ in Sub-Case 2) are relatively large: By utilizing Softmax, the hand-designed optimizer's weight becomes dominant for the $m$ parameters with relatively large gradients. This dominance guides the descent path toward $x^*$, effectively reverting to Case 1.

### B.5 PROOF FOR GRADIENT VANISHING AND EXPLODING PROPERTIES FOR THE CONTINUOUS FUNCTION IN SECTION 3.2

Recall the continuous function is represented as:

$$
f(x) = \begin{cases} 0 & x = 0 \\ \|x\|_1 (1 + \lambda \|x\|_1 + \cos \frac{1}{\|x\|_1}) & etc. \end{cases}
\tag{11}
$$

In the experiment, we use $\lambda = 0.01, d = 1000$. Since we can consider this function as a variation of $\|x\|_1$, so within in 1D:

$$
\begin{aligned}
&\textbf{Case 1: } f'(\frac{2}{(4k+1)\pi}) = 1 + \frac{4\lambda}{(4k+1)\pi} + \frac{(4k+1)\pi}{2} \underset{k \to +\infty}{\to} +\infty \\
&\textbf{Case 2: } f'(\frac{1}{(2k+1)\pi}) = \frac{2\lambda}{(2k+1)\pi} \underset{k \to +\infty}{\to} 0, \; f(\frac{1}{(2k+1)\pi}) = \frac{\lambda}{(2k+1)^2 \pi^2}
\end{aligned}
\tag{12}
$$

In equation (9), we demonstrate that gradient explosion occurs in **Case 1**, while gradient vanishing arises in **Case 2**. Additionally, we illustrate the existence of numerous local minima with low function values surrounding the global minimum of zero.

## C EXPLORATION OF VARIATIONS IN HUB STRATEGY

In this section, we will delve deeper into discussing variations of the HUB strategy, which can prove useful when dealing with more complex real-world tasks.

### C.1 INVERT WEIGHTING HUB

As mentioned earlier, VeLO's effectiveness is limited in fine-tuning tasks. Consequently, it becomes unreasonable to allocate the majority of weighting to VeLO. In the case of Inverted Weighting HUB, the following formulation is employed:

$$
U_{HUB}(g^{(t)}, \theta_H, \theta_L) = (1 - \sigma(g_l^{(t)})) \odot U_H(g^{(t)}, \theta_H) + \sigma(g_l^{(t)}) \odot U_L(g^{(t)}, \theta_L)
\tag{13}
$$

From format (5) we can see that this variation simply involves inverting the weighting matrix. This simple modification can result in a significant alteration to the behaviour of HUB, allowing for a majority of weight allocation towards the hand-designed optimizer. This approach is also preferable as it adheres to the concept of fine-tuning, where only a small number of parameters require significant adjustment to bridge the domain gap between pre-trained knowledge and downstream knowledge.

### C.2 BLOCK-WISE HUB

Another useful variation of HUB is to employ different weighting strategies in different blocks of the neural network. For instance, in many pre-trained models, a new MLP head is added to adapt to

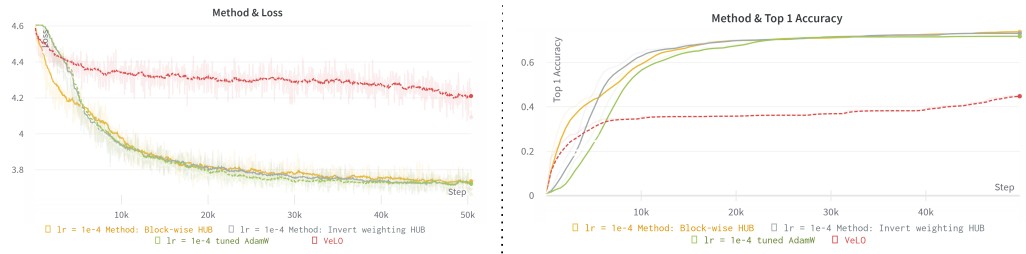

Figure 13: Using Block-wise HUB compared to Invert Weighting HUB, we can observe from the figure that both variations of HUB exhibit faster and superior performance than relying solely on one of their hybrid sources (tuned AdamW and VeLO). However, Block-wise HUB yields a slightly higher top-1 accuracy and demonstrates an evident two-stage convergence curve.

the downstream dataset, and this MLP head is trained from scratch. In such cases, it is reasonable to rely more on VeLO for fine-tuning this specific block. Therefore, we can perform a multi-stage optimization using HUB with different weighting strategies (Figure13). Specifically, for the blocks with pre-trained weights, we employ:

$$U_{HUB}(g^{(t)}, \theta_H, \theta_L) = \sigma(g_l^{(t)}) \odot U_H(g^{(t)}, \theta_H) + (1 - \sigma(g_l^{(t)})) \odot U_L(g^{(t)}, \theta_L) \tag{14}$$

Otherwise, we employ:

$$U_{HUB}(g^{(t)}, \theta_H, \theta_L) = (1 - \sigma(g_l^{(t)})) \odot U_H(g^{(t)}, \theta_H) + \sigma(g_l^{(t)}) \odot U_L(g^{(t)}, \theta_L) \tag{15}$$

### C.3 CLIPPING HUB

Clipping HUB can be represented with the following format:

$$U_{HUB}(g^{(t)}, \theta_H, \theta_L) = \sigma(g_l^{(t)}) \odot U_H(g^{(t)}, \theta_H) + (1 - \sigma(g_l^{(t)})) \odot U_L(g^{(t)}, \theta_L) \tag{16}$$

With threshold maximum = A and minimum = B

$$\text{The i-th parameter in layer l } \sigma(g_l^{(t)})_i = \begin{cases} A & \frac{exp(|g_i^{(t)}|)}{\sum_{j \in layer\ l} exp(|g_j^{(t)}|)} > A \\ B & \frac{exp(|g_i^{(t)}|)}{\sum_{j \in layer\ l} exp(|g_j^{(t)}|)} < B \\ \frac{exp(|g_i^{(t)}|)}{\sum_{j \in layer\ l} exp(|g_j^{(t)}|)} & \text{Otherwise} \end{cases} \tag{17}$$

This method proves to be particularly useful when dealing with an extremely wide network structure. As mentioned in equation (4) in section 3.2 of our paper, when the value of K approaches infinity, the HUB strategy would solely rely on VeLO, which deviates from our original intention. To address this issue, we can employ clipping HUB to prevent such a situation from occurring.

