# OpenReview forum: "HUB: Enhancing Learned Optimizers via Hybrid Update-based Strategy"
_ICLR.cc/2024/Conference — Submitted to ICLR 2024_

### Official Review · Reviewer_tuob · 2023-11-01

**Soundness:** 3 good
**Presentation:** 3 good
**Contribution:** 2 fair
**Rating:** 5
**Confidence:** 2

**Summary:**

They propose a Hybrid-Update-Based Optimization Strategy (HUB) which trade-off the involvement between a hand-designed optimizer and a learned optimizer to improve the efficient and performance. The proposed method has been validated on different types of  computer vision tasks across different types of models.

**Strengths:**

1. The method analysis is clear and visualization is helpful.
2. The experiments are solid and comprehensive.
3. The proposed method is novel and performs well on different types of  computer vision tasks across different types of models.

**Weaknesses:**

1. Lack of discussion about the computation cost of the proposed method when model size increase.
2. Lack of the GPU memory usage discussion.
3. The proposed method inspired by prompt tuning is not clear.

**Questions:**

1. Compared with other method, how will the SoftMax influence the computation time when model size increase? Since it will compute the SoftMax of all layer's gradient matrix.
2. I am curious about the GPU memory usage between the proposed method with others. Since if we want to train a large model, we will also consider if the overall pipeline can be fit on our computation resources.
3. I am still confused how the proposed method inspired by prompt tuning? I do not see the relationship between them.

---

> ### Author Response · Authors · 2023-11-14
> **Rebuttal for Reviewer (tuob)**
>
> # To Reviewer tuob
> We greatly appreciate your insightful review, which has highlighted key areas for enhancement in our work. We are addressing your concerns and suggestions as detailed below.
> ## Quick Guide:
> 1. Regarding Weakness points 1 and 2: We would like to discuss how SoftMax is much more efficient than LGL2O in terms of computational cost and memory usage.
>    1. **Memory usage：** Our method, utilizing SoftMax, does not add notable memory overhead compared to standard optimization methods. SoftMax operates on already computed gradient matrices during backpropagation, and thus, doesn't require additional memory for storing new matrices or tensors. LGL2O, on the other hand, may require more memory to store intermediate results, particularly the gradients, and parameters updated by both optimizers, which could significantly increase the memory requirement, especially for large models.
>    2. **Computational cost：** The HUB approach introduces minimal computational overhead through the SoftMax function. Each layer's time complexity is $O(N)$ (with $N$ denoting the number of parameters per layer), leading to a total complexity of $O(L⋅N)$ for $L$ layers. Since $N$ is substantially smaller than the total operations involved in training (encompassing forward and backward passes, along with other computations like batch normalization), the additional cost of HUB is akin to that of layer normalization, which also normalizes on a layer-by-layer basis. Conversely, LGL2O demands a more resource-intensive process. It requires separate model updates using both a learned optimizer and a hand-designed optimizer before each model update, followed by loss computation for the subsequent step. The choice of optimizer is then based on which one results in a lower forthcoming loss. This method essentially doubles the update steps and significantly heightens computational demands, especially in comparison to the relatively straightforward SoftMax calculation in HUB.
> 2. Regarding Weakness point 3: Our method is inspired by the concept of prompt tuning, wherein both strategies aim to maximize the latent potential of pre-trained models without necessitating extensive additional training. Prompt tuning typically guides a pre-trained model to adapt to new tasks or data by incorporating human-understandable rules and templates into the input, thereby unlocking the model's inherent capabilities. Similarly, the HUB strategy enhances a learned optimizer's performance by directing it with hand-designed rules. This approach aligns with our overarching goal of optimizing the utilization of existing models and optimizers, efficiently and effectively, without demanding significant extra resources.
>
> Your feedback is invaluable in refining our work, and we hope this response adequately addresses your points, offering enhanced clarity and depth to our submission.
> ## References
> [A] Pr'emont-Schwarz, Isabeau et al. “A Simple Guard for Learned Optimizers.” ArXiv abs/2201.12426 (2022): n. pag.

---

### Official Review · Reviewer_i8fH · 2023-11-05

**Soundness:** 2 fair
**Presentation:** 3 good
**Contribution:** 1 poor
**Rating:** 3
**Confidence:** 4

**Summary:**

Learned optimizers play a crucial role in meta-learning, offering improved performance compared to traditional, manually designed optimizers across various tasks. However, these learned optimizers face challenges such as handling tasks beyond their training data, unpredictable behavior, and suboptimal performance in fine-tuning tasks. The authors’ introduce a Hybrid-Update-Based (HUB) optimization strategy that integrates hand-designed and learned optimizers. HUB is tested on a variety of tasks to illustrate its efficacy in out-of-distribution tasks. The authors also provide some theoretical analysis of HUB.

**Strengths:**

- Improving the working of learned optimizers (LO) is a challenging and important task to increase their adoption. The problem addressed in the paper is thus relevant and important to the community.
- A key strength of the approach is its simplicity: a convex combination of the LO and hand-designed update. The approach is significantly simpler and more efficient than LGL2O that requires actually checking the model performance after the updates from both Lo and hand-designed optimizer. This combination weight is derived from the softmax of the gradients.
- Experiments are adequate both in quality and quantity. I especially like the experiments on out-of-distribution tasks and fine-tuning.

**Weaknesses:**

- How does one reconcile with the use of a hand-designed optimizer during training when the goal is to train using a robust LO? Doesn’t it defeat the purpose of an LO? Strategies that enable LO to work well (stable convergence) will be more interesting and beneficial. The ideas presented in the paper do not improve the robustness of the LO, rather mitigate issues when using it for training a different task. The results indicate only a marginal difference between VeLO (LO), ADAM, and HUB results. Thus, begging the question, the need for an LO.
- I find the motivation driven by vanishing gradients and the hybrid update equation 5 to be contradicting. When faced with vanishing gradients the convex combination factor of the hand-designed update will be close to 0, forcing the updates to depend on the LO. As LO can be unstable near the optima, how does the proposed strategy facilitate model convergence?
- There has been (at least one) recent work [1] that also attempts to overcome the challenges with LO raised in the paper. This method appears to be much simpler than the proposed approach. It would be helpful to have some comparison against it.
- Can you clarify if the stability results on ADAM are authors’ contributions or a restatement of existing literature?
[1] Learning to learn with better convergence, P Chen, S Reddi, S Kumar and C-J, Hsieh.

**Questions:**

please see the weaknesses

--- Post authors' response
I thank the authors for the detailed response.

While I agree that the approach is simple, I am not entirely convinced with the motivation  of combining LO and SGD updates. Hence, I retain my recommendation.

---

> ### Author Response · Authors · 2023-11-13
> **Rebuttal for Reviewer (i8fH)**
>
> # To Reviewer (i8fH)
> Thanks for the feedback, we appreciate your time and effort to evaluate our work. We will address your concerns and suggestions in the following sections.
> ## Quick Guide:
> 1. Regarding Weakness point 1: In our opinion, the primary objective of training is always to achieve optimal performance within the allocated time constraints. Therefore, there should be no prohibition on utilizing either LO or hand-designed optimizers, or even a combination of both, as long as they contribute towards achieving this goal. Prior works such as GL2O and LGL2O have been accepted by the community, indicating a recognition of the value in leveraging a combination of approaches. Importantly, the incorporation of human-guided rules into deep neural networks is a common practice in various domains, including NLP and CV (e.g., prompt tuning). Our proposed method provides a plug-and-play solution for effectively integrating human rules to guide learned optimizers. Furthermore, based on experimental results in Sections 3.2 and 4.1, we believe that we provide strategies enabling LO to exhibit stable convergence, as previously mentioned. Notably, when dealing with tasks in distribution settings, VeLO demonstrates significantly faster convergence compared to tuned Adam and AdamW (refer to Figure 2(c), Figure 4(b), Figure 5, and Figure 12).
> 2. Regarding Weakness point 2: As you correctly stated, 'learned optimizer can be unstable near optima', it also implies some times learned optimizer is stable. In fact, we have demonstrated in Section 3.2 and 4.2 that VeLO can achieve fast and stable convergence in in-distribution settings without any guidance. So the rationale behind our methods is to give chance for VeLO to explore the parameter space and then use the hand-designed optimizer to stabilize the training process once it deviates too far. This process may face fluctuations, as shown in Section 3.2, but compared to other competitors, we have shown that HUB can better facilitate convergence.
> 3. Regarding Weakness point 3 and 4: The work you provided is not directly comparable to ours, as they discuss how to train a more powerful learned optimizer from scratch. In contrast, our method aims to improve the stability and generalization ability of a pre-trained learned optimizer in a plug-and-play manner, without additional training. Furthermore, our baseline VeLO already adopts a layer-wise LSTM structure, which is similar to the coordinate-wise RNN structure proposed in this paper. The stability results on ADAM have already been proven in the original paper on Adam, and here, we provide a concise proof in Sec B.2 in the Appendix. Our primary purpose in proving this is to offer a better understanding of the stability of hand-designed optimizers.
>
> In conclusion, we believe that our method is a simple yet effective solution for enhancing the stability and generalization ability of learned optimizers. We hope that our response has addressed your concerns.
> ## References
> [A] Metz, Luke et al. “VeLO: Training Versatile Learned Optimizers by Scaling Up.” ArXiv abs/2211.09760 (2022): n. pag.
>
> [B] Chen, Reddi et al. "Learning to learn with better convergence"
>
> [C] Kingma, Diederik P. and Jimmy Ba. “Adam: A Method for Stochastic Optimization.” CoRR abs/1412.6980 (2014): n. pag.

---

### Official Review · Reviewer_BBSt · 2023-11-09

**Soundness:** 2 fair
**Presentation:** 2 fair
**Contribution:** 2 fair
**Rating:** 5
**Confidence:** 4

**Summary:**

This paper presents a simple algorithm to prevent learned optimizers from collapsing (e.g., giving incorrect outputs). The proposed algorithm is essentially a weighted average between learned and hand-crafted optimization rules. On multiple tasks, both in- and out-of-distribution, the authors show that the proposed algorithm can help boost performance.

**Strengths:**

- The topics is interesting to me. Learned optimization could be a promising topics for future research.
- The proposed techniques are simple and straight-forward to implement.

**Weaknesses:**

A few relevent papers are missing from the reference which I will explain as follows:

[r1] distills the learned rules into specific mathematical expressions, which also provides "effective control". There are also works characterizing the generalization behaviors of L2O [r2-r4]. Please consider discuss these works.

[r1] Symbolic Learning to Optimize: Towards Interpretability and Scalability

[r2] Hyperparameter tuning is all you need for lista

[r3] Understanding deep architecture with reasoning layer

[r4] M-L2O: Towards Generalizable Learning-to-Optimize by Test-Time Fast Self-Adaptation

-----

Regarding Figure 2(c), would it be an easier solution to use just early stopping?

-----

There is almost no analysis to experiments results in Section 4.1. Section 4.2 paragraph 1 seems to be cut off and unfinished. Also I have a same question here: would the baselines experience fluctuations? Can early stop can help?

-----

I would not criticize this paper as lacking novelty as the authors have already conducted a lot of experiments. However I think there is plenty of rooms for the presentation to be improved. Spacing could be make more compact to accommodate more details regarding the experiments. Right now I have to look back and forth to understand what the tasks really are.

-----

Minor errors:
"tasks undertaken during training", Traditional optimizers such as SGD", "convext optimization problems": a space is missing before the reference. Also the whole contribution bullet #2. Please proof-read carefully and add necessary spacing.

**Questions:**

"Consequently, even a few misguided steps within this “black box” can significantly disrupt the entire training process, with no means to prevent such adversities." -> is there any example to support this?

---

> ### Author Response · Authors · 2023-11-13
> **Rebuttal for Reviewer (BBSt)**
>
> # To Reviewer (BBSt)
> Thank you for your meticulous and invaluable feedback. We genuinely appreciate the time and effort you dedicated to evaluating our work, and we welcome this opportunity to clarify and enhance our study based on your input.
> ## Quick Guide:
> 1. Regarding Weakness Point 1: While we acknowledge the effectiveness of alternative approaches in controlling learned optimizers, our methodology, as extensively outlined in the entire second paragraph of our introduction, primarily focuses on addressing the specific challenges associated with fine-tuning or training a learned optimizer. Our method and baselines are designed to be easily integrated and offer simplicity and effectiveness at a lower implementation cost. This provides us with a distinct advantage over some of the methods you've mentioned, which will be further elaborated on below in Detailed Discussion.
> 2. Regarding Weakness Point 2: With regards to your suggestion of utilizing early stopping, it is indeed a widely employed technique in practical scenarios. However, for all the experiments conducted in Section 4, we have already compared against the historical best evaluation checkpoint, which surpasses the effectiveness of simple early stopping. Moreover, Section 3.2 primarily focuses on employing a setup function to demonstrate how HUB's negative feedback attribute offers enhanced control over learned optimizers even during extended training steps. Our ultimate objective for this experiment renders it unsuitable for implementing early stopping.
> 3. Regarding Weakness Points 3, 4, and 5: We would like to apologize for the omission of a full stop sign after Section 4.2, paragraph 1. We assure you that we will thoroughly review this issue along with other typographical errors, inappropriate notations, and spacing inconsistencies in order to rectify them before the camera-ready version is submitted. Furthermore, we would like to clarify that Sections 4.1 and 4.2 primarily aim to provide a comprehensive overview and empirical evidence of the effectiveness of our methodology at a high level. As mentioned at the beginning of Section 4, for an extensive and detailed analysis of the experimental setup, methodology, and results, please refer to Section A in the appendix.
> 4. Regarding Question Point 1: Yes, we can certainly provide some examples. In the technique report of VeLO, specifically in Section 4.4 titled "Limitations and Failure Cases," the author enumerates four prevalent failure scenarios. By analyzing the loss curve presented in these cases, it becomes evident that failures often manifest during the middle of training with abrupt fluctuations in loss values. Subsequently, this initial fluctuation leads to increased instability and eventual failure of the training process. Remarkably similar to our demonstrated failed case in Section 4.1 paragraph 2 when dealing with unseen LTC architecture, a sudden loss fluctuation occurs followed by NAN.

---

> > ### Author Response · Authors · 2023-11-13
> > **Detailed Dicussion for Weakness Point 1:**
> >
> > As discussed briefly in the Quick Guide, our methodology places a strong emphasis on minimizing tryout costs, a crucial attribute for practical applications of learned optimizers. While we acknowledge in the last section of our paper that more robust solutions may be achieved through guided training, we aim to highlight the unique contributions of our research through the following comparisons:
> > 1. Symbolic Learning to Optimize: This method introduces a symbolic representation and interpretation framework for Learning to Optimize (L2O), with the goal of enhancing scalability and interpretability. Although its focus on symbolic regression for creating interpretable and scalable models is valuable, it does not directly address the ease of fine-tuning existing models and adaptability to varying task distributions, as offered by our method. Additionally, the training process of symbolic regression, while more efficient than RNN-based learned optimizers, can still be computationally complex. For instance, meta-training a symbolic optimizer on ResNet-50 requires approximately 30GB of GPU memory and 200 epochs. In contrast, our method provides a straightforward combination of learned and hand-crafted rules, making it computationally less intensive and more easily implementable in diverse environments.
> > 2. HyperLISTA: HyperLISTA streamlines the training of LISTA by tuning only three hyperparameters and achieving superlinear convergence. However, this method is tailored to LISTA networks and may not be as broadly applicable as our approach. Our method consistently demonstrates performance improvements across various neural network architectures (as detailed in Section 4) without the need for hyperparameter tuning specific to any particular architecture.
> > 3. Understanding Deep Architectures with Reasoning Layer: This paper focuses on integrating deep learning models with reasoning, emphasizing trade-offs in algorithm properties such as convergence, stability, and sensitivity. While their approach contributes to a theoretical understanding of hybrid architectures, our method specifically addresses the practical application of learned optimizers with human rules guidance in a wide range of tasks.
> > 4. M-L2O showcases rapid task adaptation in out-of-distribution tasks but relies on a nested structure and meta-training, which could be complex to implement in practice. In contrast, our approach can be readily applied to a wider array of tasks without requiring such an extensive setup, as demonstrated in our experiments with diverse datasets.
> >
> > In conclusion, while the cited methods contribute significantly to the field, our approach distinguishes itself through its ease of implementation and broad applicability, offering a pragmatic solution in resource-constrained environments.
> > ## References
> > [A] Metz, Luke et al. “VeLO: Training Versatile Learned Optimizers by Scaling Up.” ArXiv abs/2211.09760 (2022): n. pag.
> >
> > [B] Zheng, Wenqing et al. “Symbolic Learning to Optimize: Towards Interpretability and Scalability.” ArXiv abs/2203.06578 (2022): n. pag.
> >
> > [C] Chen, Xiaohan et al. “Hyperparameter Tuning is All You Need for LISTA.” Neural Information Processing Systems (2021).
> >
> > [D] Chen, Xinshi et al. “Understanding Deep Architectures with Reasoning Layer.” ArXiv abs/2006.13401 (2020): n. pag.
> >
> > [E] Yang, Junjie et al. “M-L2O: Towards Generalizable Learning-to-Optimize by Test-Time Fast Self-Adaptation.” ArXiv abs/2303.00039 (2023): n. pag.

---

### Official Review · Reviewer_dXMr · 2023-11-13

**Soundness:** 2 fair
**Presentation:** 1 poor
**Contribution:** 2 fair
**Rating:** 5
**Confidence:** 4

**Summary:**

This work proposed a hybrid scheme for training neural networks, which combines classic, hand-designed optimizers (e.g., Adam) and learned optimizers (e.g., VeLO). The proposed hybrid scheme is a reweighted combination of the updates from a hand-designed optimizer and a learned optimizer. The reweighting coefficients are decided by the gradient magnitudes of weights within one each layer, by feeding them into a softmax function. The formulation in Eqn. (5) implied that the proposed hybrid scheme mostly depends on the learned optimizer updates, with very weak influence from the hand-designed updates, while the authors suggested that the coefficients can be inverted for specific tasks like fine-tuning.

The authors conducted experiments on a wide range of training tasks and neural networks.

**Strengths:**

+ The integration of learned optimizers with classic, hand-designed optimizers has been desired for a long time.
+ The proposed method is simple enough and easy to implement.
+ The authors did a large amount of empirical experiments to show the effectiveness of the proposed method.

**Weaknesses:**

- The proposed method is a hybrid scheme. A key issue of a hybrid scheme is the difficulty to guarantee the convergence, which is the highlight of the two highly-related methods mentioned in the paper: (Pr'emont-Schwarz et al., 2022; Heaton et al., 2020). The authors mentioned the computation complexity but discard the discussion of theoretical analysis.

- This paper is poorly written (to the extent where the overall quality of this work is significantly influenced in my humble opinion), making it difficult to understand the intuition behind the default hybrid scheme formulated in Eqn (5). The arrangement of contents also made it difficult to read, such as the tables in the experiment section.

- Considering the essense of high-demensionality of modern neural networks, the influence brought in by the hand-designed update in Eqn. (5) would be really weak. Probabily only one or two weights in each layer that have really dominant gradient will lean towards the hand-designed update in a meaningful sense. The authors did not show the histogram of the elements of the weighting matrix throughout the manuscript. I am really curious to see such observations to understand the behaviors of the proposed method in real-world scenarios.

**Questions:**

- In Fig. 1, the authors illustrated that LGL2O calculates the loss and gradient for the two types of optimizers separately. Are there specific technical designs that keep us from using the same shared computation strategy as in HUB?

- The baseline accuracies of ResNet-50 and Xception on CIFAR-10 are too low. I checked the benchmark on CIFAR-10. These two networks should be able to easily achieve >95% testing accuracies. I wonder if the baseline networks are correctly and fully trained.

---

> ### Author Response · Authors · 2023-11-20
> **Rebuttal for Reviewer (dXMr)**
>
> # To Reviewer (dXMr)
> First, we greatly appreciate receiving such a valuable comment with a clear and specific description of your concerns. It gives us a sense of accomplishment and we are committed to addressing the weak point you have pointed out in order to make a more solid contribution to the community.
> ## Quick Guide:
> 1. Regarding Weakness point 1: In brief, We would offer a more theoretical elucidation of the computational complexity of HUB in Part 1 of our response. Additionally, we apologize for seeking clarification, but it is imperative for us to ensure that we comprehensively understand and accurately explain your concern. Based on our best knowledge, neither a hand-designed optimizer nor a learned optimizer can guarantee unconditional convergence. Besides, in LGL2O, they already proved conditional convergence can be guaranteed with a hybrid scheme. Hence, within Part 1 of our response, we intend to dissect this question into various scenarios and provide an elaborate explanation.
> 2. Regarding Weakness point 2: We apologize for the compact table arrangement due to page limitations, which may cause confusion. However, we made efforts to ensure that the tables are placed near their corresponding paragraphs for easy reference. Additionally, we are pleased to note that two other reviewers have given positive ratings on the presentation of our paper. Nevertheless, we understand that subjective opinions can vary and would greatly appreciate more specific suggestions on improving any unclear aspects of our presentation.
> 3. Regarding Weakness point 3: In Sec. C of our supplementary material, we have discussed three types of variations of HUB, and one of the variations - the Clipping HUB sets a bound for the hybrid ratio to avoid allocating very small weights to one of the optimizers. We find that this variation can be useful when the network is extremely wide/narrow. However, the improvement is marginal because the weight allocation is dynamically adjusted during training, we would update the figure in the rebuttal PDF to show the result of the variations of HUB.
> 4. Regarding Question point 1: As you correctly stated, in order to reduce cost, HUB uses a shared gradient matrix for computing the hybrid ratio. However, this is impossible in LGL2O, because the main idea for LGL2O is to refer to future loss. In other words, LGL2O needs to compute the gradient matrix for both optimizers and then compute the loss for both optimizers, and then compare the loss to decide which optimizer to use. If we try to share the gradient matrix, the loss will be the same for both optimizers, and the comparison will be meaningless.
> 5. Regarding Question point 2: As mentioned in Sec. A of our supplementary material, we used the pre-trained model Xception and scratch model ResNet-50 for the JAX framework from our codebase https://github.com/abarcel/haikumodels. This codebase gives a basic implementation of the models and the results are matched to the original ***Deep Residual Learning for Image Recognition*** paper (https://paperswithcode.com/paper/deep-residual-learning-for-image-recognition). The >95% Top1 accuracy you mentioned is probably using the data-augmentation technique shown in ***ResNet strikes back: An improved training procedure in timm***(https://paperswithcode.com/paper/resnet-strikes-back-an-improved-training).

---

> > ### Author Response · Authors · 2023-11-20
> > **Detailed Dicussion for Weakness Point 1:**
> >
> > The HUB approach introduces minimal computational overhead through the SoftMax function. Each layer's time complexity is $O(N)$ (with $N$ denoting the number of parameters per layer), leading to a total complexity of $O(L⋅N)$ for $L$ layers. Since $N$ is substantially smaller than the total operations involved in training (encompassing forward and backward passes, along with other computations like batch normalization), the additional cost of HUB is akin to that of layer normalization, which also normalizes on a layer-by-layer basis. Conversely, LGL2O demands a more resource-intensive process. It requires separate model updates using both a learned optimizer and a hand-designed optimizer before each model update, followed by loss computation for the subsequent step. The choice of optimizer is then based on which one results in a lower forthcoming loss. This method essentially doubles the update steps and significantly heightens computational demands, especially in comparison to the relatively straightforward SoftMax calculation in HUB.
> >
> > Hand-designed optimizers like Adam can only ensure convergence in convex optimization problems (we provide a brief proof of this property in Sec B.2 in the Appendix). However, deep learning optimization typically occurs within high-dimensional non-convex spaces where no optimization method can guarantee convergence reliably. In this case, I assume your concern could be rephrased as "Hybrid scheme is the difficulty to guarantee the convergence under convex optimization problems." To address this question effectively:
> >
> > **Regarding hand-designed optimizers:** As mentioned earlier, hand-designed optimizers generally ensure global convergence in convex optimization problems due to their gradient descent mechanism based on mathematical rules such as Momentum. The updating process is transparent and explainable.
> >
> > **Regarding learned optimizers:** Convergence cannot be guaranteed for learned optimizers, given that their functioning is vastly different from traditional optimizers—they learn how to update parameters by training on a multitude of tasks and this can be influenced by many factors, including the quality and quantity of training data, as well as the design and training process of the optimizer. Therefore, convergence theories from traditional optimizers cannot be directly applied and the updating process of the learned optimizer remains opaque and can be considered as a black box.
> >
> > Although learned optimizers like VeLO cannot guarantee convergence, this does not necessarily imply inferiority to hand-designed optimizers. In fact, in real-world optimization tasks, it endows the learned optimizer with a stronger ability to escape saddle points. We have also discussed this in Sec. 3.2 in our paper. For a more detailed comparison between VeLO and hand-designed optimizers, please refer to Figure 1 of the original VeLO technique report[C]. HUB is a hybrid method that combines two types of optimizers. Just like the convergence-proving method shown in LGL2O, we can also prove the convergence of HUB under convex optimization problems by the same strategy. This is listed in the Sec. B.4 of the Appendix. We hope that our response has addressed your concerns.
> > ## References
> > [A] Metz, Luke et al. “VeLO: Training Versatile Learned Optimizers by Scaling Up.” ArXiv abs/2211.09760 (2022): n. pag.
> >
> > [B] Pr'emont-Schwarz, Isabeau et al. “A Simple Guard for Learned Optimizers.” ArXiv abs/2201.12426 (2022): n. pag.
> >
> > [C] He, Kaiming et al. “Deep Residual Learning for Image Recognition.” 2016 IEEE Conference on Computer Vision and Pattern Recognition (CVPR) (2015): 770-778.
> >
> > [D] Wightman, Ross et al. “ResNet strikes back: An improved training procedure in timm.” ArXiv abs/2110.00476 (2021): n. pag.

---

### Comment · Area_Chair_FxMB · 2023-11-23
**From AC at the end of rebuttal: Reviewer response required**

Dear Reviewers,

Thanks for your time and commitment to the ICLR 2024 review process.

As we approach the conclusion of the author-reviewer discussion period (Wednesday, Nov 22nd, AOE), I kindly urge those who haven't engaged with the authors' dedicated rebuttal to please take a moment to review their response and share your feedback, regardless of whether it alters your opinion of the paper.

Your feedback is essential to a thorough assessment of the submission.

Best regards,

AC

---

### Meta-Review · Area_Chair_FxMB · 2023-12-10

**Metareview:**

This paper presents a nice integration between hand-designed optimizers and learned optimizers, which is shown to significantly reduce the computational overhead and enhance the generalization to out-of-distributions. After rebuttal and discussion, most of the concerns raised were addressed as acknowledged by the reviewers. However, three important concerns remained. One is the questionable motivation of integrating hand-designed and learned optimizers. While the community had seen some L2O methods, they were not widely adopted by the community, not to say the hybrid optimizers. Second, the empirical performance is not superior in general to hand-designed solvers, given the more cumbersome hybridization nature. Third, theoretical guarantee is missing even for the convex optimization case, which is not acceptable, because such an understanding is important to make an optimization approach (recall SGD, Adam, etc) better grounded and improvable by upcoming research. Based on these considerations, the paper is rejected in its current form.

**Justification For Why Not Higher Score:**

Three major flaws prevent the paper from acceptance, which are weak motivation of optimizer hybridization, no theoretical guarantees and uncompetitive performance.

**Justification For Why Not Lower Score:**

N/A

---

### Decision · Program_Chairs · 2024-01-16

Reject